# A Phase Ib Expansion Cohort Evaluating Aurora A Kinase Inhibitor Alisertib and Dual TORC1/2 Inhibitor Sapanisertib in Patients with Advanced Solid Tumors

**DOI:** 10.3390/cancers16081456

**Published:** 2024-04-10

**Authors:** S. Lindsey Davis, Wells A. Messersmith, W. Thomas Purcell, Elaine T. Lam, Bradley R. Corr, Alexis D. Leal, Christopher H. Lieu, Cindy L. O’Bryant, Stephen G. Smoots, Evan D. Dus, Kimberly R. Jordan, Natalie J. Serkova, Todd M. Pitts, Jennifer R. Diamond

**Affiliations:** 1Division of Medical Oncology, Department of Medicine, University of Colorado School of Medicine, Aurora, CO 80045, USA; 2Division of Hematology and Oncology, Department of Medicine, University of Washington School of Medicine, Seattle, WA 98195, USA; 3Division of Gynecologic Oncology, Department of Obstetrics and Gynecology, University of Colorado School of Medicine, Aurora, CO 80045, USA; 4Skaggs School of Pharmacy and Pharmaceutical Sciences, Aurora, CO 80045, USA; 5Department of Immunology and Microbiology, University of Colorado School of Medicine, Aurora, CO 80045, USA; 6Department of Radiology, University of Colorado School of Medicine, Aurora, CO 80045, USA

**Keywords:** alisertib, sapanisertib, solid tumors, pancreatic adenocarcinoma, expansion cohort, Aurora A kinase, mTOR

## Abstract

**Simple Summary:**

Drugs that target Aurora A kinase as a treatment for cancer can lead to activation of the PI3K/AKT/mTOR pathway and ultimately resistance to treatment. We evaluated the preliminary efficacy of the addition of an inhibitor of this pathway (TORC1/2 inhibitor sapanisertib) to Aurora A kinase inhibitor alisertib in patients with advanced solid cancers, including a group with pancreatic adenocarcinoma. Select patients experienced prolonged stabilization of their cancers when treated with this combination therapy, including two with pancreatic cancer. To explore potential differences in response to this therapy, changes in the tumor tissues and immune system were evaluated, as were changes in imaging that can evaluate tumor activity. The genetic make-up of the tumor as well as spread of cancer to the liver may contribute in this setting.

**Abstract:**

Background: This study further evaluated the safety and efficacy of the combination of alisertib and sapanisertib in an expansion cohort of patients, including a subset of patients with refractory pancreatic adenocarcinoma, with further evaluation of the pharmacodynamic characteristics of combination therapy. Methods: Twenty patients with refractory solid tumors and 11 patients with pancreatic adenocarcinoma were treated at the recommended phase 2 dose of alisertib and sapanisertib. Adverse events and disease response were assessed. Patients in the expansion cohort were treated with a 7-day lead-in of either alisertib or sapanisertib prior to combination therapy, with tumor tissue biopsy and serial functional imaging performed for correlative analysis. Results: Toxicity across treatment groups was overall similar to prior studies. One partial response to treatment was observed in a patient with ER positive breast cancer, and a patient with pancreatic cancer experienced prolonged stable disease. In an additional cohort of pancreatic cancer patients, treatment response was modest. Correlative analysis revealed variability in markers of apoptosis and immune cell infiltrate according to lead-in therapy and response. Conclusions: Dual targeting of Aurora A kinase and mTOR resulted in marginal clinical benefit in a population of patients with refractory solid tumors, including pancreatic adenocarcinoma, though individual patients experienced significant response to therapy. Correlatives indicate apoptotic response and tumor immune cell infiltrate may affect clinical outcomes.

## 1. Introduction

Aurora A kinase is essential to mitotic regulation and is overexpressed in many cancers [1,2]. Accordingly, drugs targeting Aurora A kinase are of significant interest as cancer therapy and have been evaluated in multiple clinical settings [3,4]. Alisertib, the most extensively studied of the Aurora A kinase inhibitors, is an oral, selective small molecule inhibitor of Aurora A kinase [5], and has an FDA orphan drug designation for small cell lung cancer [6]. Efficacy as a single-agent has been relatively limited in other tumor types, and approaches evaluating combination therapy are being explored to combat resistance [5,7].

Activation of the PI3K/AKT/mTOR pathway has long been associated with resistance to multiple targeted therapies across multiple tumor types [8,9,10], leading to the evaluation of combination therapies integrating inhibitors of this pathway. Sapanisertib is a next-generation oral small-molecule mTOR inhibitor targeting TORC1 and TORC2 [11,12] that has been evaluated in combination with both targeted therapies and cytotoxic chemotherapy in multiple clinical trials.

In our prior work, senescence and upregulation of genes in the PI3K/AKT/mTOR pathway were observed in triple-negative breast cancer (TNBC) patient-derived xenograft models treated with alisertib to resistance, and tumor growth inhibition was demonstrated in those models treated with alisertib and sapanisertib in combination [13]. This has been confirmed by others in TNBC models [14], and has been linked to suppression of autophagic cell death by Aurora A kinase, which is facilitated by mTOR activation in TNBC [15].

Given these findings, combination therapy with alisertib and sapanisertib was evaluated in patients with advanced solid tumors in a phase 1 dose-finding study [16]. In this trial, combination therapy was determined safe and tolerable at a recommended phase 2 dose (RP2D) of alisertib 30 mg BID on days 1–7 of a 21-day cycle and sapanisertib 2 mg daily on a continuous schedule. Most common adverse events related to treatment in this population were neutropenia, fatigue, nausea, mucositis and rash. Most of these events were mild, though grade 3 and 4 neutropenia and fatigue were observed. In this population, one patient with hormone receptor positive, HER2 negative (HR+/HER2−) breast cancer and one with castrate-resistant prostate cancer experienced prolonged stable disease with study treatment.

The purpose of this study was to further evaluate safety, as well as preliminary efficacy, in an expansion cohort of patients treated at the RP2D, and to further explore the pharmacodynamics of the combination in human patients through correlative assessments on serial tumor tissue samples and functional imaging.

## 2. Materials and Methods

### 2.1. Patient Selection

Patients were assigned to one of two cohorts, the solid tumor expansion cohort or the pancreatic cancer expansion cohort. In the solid tumor expansion cohort, patients with a histologically confirmed solid tumor that was incurable or refractory to standard therapy, or for which no standard therapy exists, were eligible to participate. The purpose of this cohort was to better understand the safety and efficacy of the combination of alisertib and sapanisertib in this patient population, and to further explore the pharmacodynamic effects of the treatment. These patients were required to have at least one tumor lesion amenable to repeat core needle or punch biopsy for correlative testing. Due to the clinical benefit noted in pancreatic cancer patients treated in this cohort as well as the prior dose-finding study [16], an additional pancreatic cancer expansion cohort was added to further explore efficacy in this specific population. In this cohort, patients were required to have a histologically confirmed diagnosis of locally advanced or metastatic pancreatic adenocarcinoma, which was refractory to standard therapy.

Patients in all cohorts were required to be greater than 18 years old and have an ECOG performance status of 0 or 1. Adequate hematologic, hepatic, renal, and cardiac function were required, as were hemoglobin A1c of less than 7% and fasting triglycerides less than or equal to 300 mg/dL. Patients with treated brain metastases who no longer required steroids or anti-epileptic drugs and had no evidence of progression after treatment were eligible to participate. Patients who required supplemental oxygen or had a condition that could result in excessive daytime sleepiness were excluded, as were patients with a condition that could alter the absorption of the study medications.

The protocol was approved by the local institutional review board, and all patients signed a written consent prior to enrollment according to federal and institutional guidelines.

### 2.2. Study Design

Eligible patients in both cohorts were treated with the combination of alisertib 30 mg by mouth twice daily on days 1–7 in a 21-day cycle and sapanisertib 2 mg by mouth daily according to the previously defined RP2D [16]. In Cycle 1 of the solid tumor expansion cohort only, patients were assigned to treatment with either alisertib or sapanisertib as a single-agent on days 1–7 (Figure 1). For the remainder of the study, these patients received combination treatment. Patients in both the alisertib and sapanisertib lead-in groups were eligible to participate in biopsy and imaging correlative studies. In the pancreatic cancer expansion cohort, patients were treated with both agents at the RP2D without a single-agent lead-in (Figure 1). These patients were not included in correlative studies.

Patients in both cohorts were instructed to complete daily glucose monitoring on a provided glucometer at home after fasting overnight for a minimum of 8 h, with data reviewed at each study visit. Hyperglycemia observed during home glucose monitoring was confirmed in the clinic.

With each treatment cycle, initiation of alisertib was delayed for grade 2 or greater neutropenia or thrombocytopenia, or other grade 2 or greater treatment related toxicities that had not resolved from a prior treatment cycle. Alisertib was reduced by one dose level for grade 4 or symptomatic anemia or thrombocytopenia, or grade 3 non-hematologic toxicity, including nausea, vomiting, or diarrhea that persisted despite optimal supportive care. Study treatment with sapanisertib was withheld in the setting of grade 3 or greater treatment-related toxicities, despite optimal supportive care. Up to two dose level reductions of sapanisertib were allowed for those patients with grade 3 events that resolved within two weeks.

Disease status was assessed by imaging using RECIST 1.1 at baseline and after every 3 cycles of treatment. Objective responses were confirmed on repeat evaluation at least 4 weeks after the initial documentation.

Study treatment was discontinued in the setting of disease progression, unacceptable toxicity, or protocol non-compliance. Treatment with either alisertib or sapanisertib alone was allowed if either drug was discontinued due to unacceptable toxicity.

### 2.3. Correlative Studies

Correlative studies were performed in a subset of patients enrolled in the solid tumor expansion cohort as described below.

#### 2.3.1. Functional Imaging

T2 weighted MRI (T2w MRI) and diffusion weighted imaging (DWI) were performed to assess structural changes and tissue cellularity, respectively, in a subset of patients with confirmed hepatic metastases. All MRI scans were acquired on a Siemens Skyra 3 Tesla MRI scanner (Siemens Healthineers, Malvern, PA, USA) equipped with a 32-channel phased-array body coil. The entire liver was included in the field of view (38 × 38 cm^2^). The MRI protocol consisted of a conventional axial fast spin echo T2 weighted imaging (FSE T2WI) sequence, followed by an axial spin echo echoplanar imaging (SE-EPI) DWI sequence with multiple b-values (0, 100, 200, 600, 800, 1000, 1200 s/mm^2^). Total tumor burden (reported in cm^3^) was calculated from the sequential T2w MRI and compared with the apparent diffusion coefficient (ADC) values from DWI as an assessment of tumor cellularity and apoptosis in order to determine its function as a potential biomarker of early treatment response [17,18]. DWI and T2w MRI were performed prior to treatment initiation, C1D7 (after single-agent lead-in), and C2D7 (after combination treatment) (Figure 1).

#### 2.3.2. Tumor Tissue Evaluation

Biopsies were performed in patients in the solid tumor expansion cohort prior to treatment initiation, Cycle 1 Day 7 (C1D7) (after single-agent lead-in), and Cycle 2 Day 7 (C2D7) (after combination treatment) (Figure 1) to assess the pharmacodynamic effects of treatment. Fresh tissue samples from patients in both treatment groups at each timepoint were divided, with portions of formalin fixed and paraffin embedded, treated with O.C.T. compound, and flash frozen.

##### Fluorescence Microscopy

FFPE tissues were sectioned by the Pathology Shared Resource (RRID: SCR_021989). Multi-spectral imaging was performed by the Human Immune Monitoring Shared Resource (RRID: SCR_021985) as previously described [19]. Sequential staining was performed on tissues according to each of the following 3 treatment panels: Panel 1) DAPI, HuCK (Dako, Glostrup, Denmark, catalogue number M3515), Hup53 (Leica, Wetzlar, Germany, sc-6243), HuCyclin B1 (abcam, Cambridge, UK, ab32053), HuKi67 (EPREDIA (Thermofisher, Waltham, MA, USA), RM-9106-S), Hup21 (abcam, Cambridge, UK, ab109520), HuHH3 (abcam, ab1220); Panel 2) DAPI, HuCK (Dako, M3515), HuCD45 (Leica, PA0042), HuCl Casp 3 (Cell Signaling Technology, Danvers, MA, USA, 9664L), HupS6 (Cell Signaling Technology, 211S); Panel 3) DAPI, HuCK (Dako, M3515), HuCD3 (Leica, PA0553), HuCD4 (Biocare, Pacheco, CA, USA, API3209AA), HuCD19 (Leica, PA0843), HuCD56 (Leica, PA0191), HuCD8 (Dako, M7103), HuFOXP3 (abcam, AB20034), HuCD68 (Dako, M0814). Six color multi-spectral imaging was performed for Panels 1 and 2 using the Perkin Elmer Vectra 3 instrument, and for Panel 3 using the Vectra Polaris (Phenolmager HT) instrument (Akoya Biosciences, Marlborough, MA, USA). Briefly, the slides were deparaffinized, heat treated in antigen retrieval buffer, blocked, and incubated with primary antibody, followed by horseradish peroxidase (HRP)-conjugated secondary antibody polymer, and HRP-reactive OPAL fluorescent reagents. The slides were stripped in between each stain with heat treatment in antigen retrieval buffer. Whole slide scans were collected and multi-spectral images of each tissue were then collected using the 20× objective with a 0.5-micron resolution.

For quantification, images were first spectrally unmixed using references collected on the Vectra 3.0 or PhenoImager HT, respectively, and an unstained control reference was used to subtract auto-fluorescence in inForm software (version 2.5 and 2.6, respectively). Tissues were segmented into tumor regions (CK+/DAPI+), non-tumor tissue (CK-/DAPI+), and glass (CK-/DAPI-/auto-fluorescence-). For Panels 1 and 2 collected on the Vectra 3 microscope, single cells were segmented using DAPI nuclear staining (relative intensity 0.25) and assisted with CK cytoplasmic staining. The minimum nuclear size was set to 14 pixels with a splitting sensitivity of 0.4. For Panel 3 collected on the PhenoImagerHT, the relative DAPI intensity was set to 0.38 and nuclear splitting was assisted with CK and CD68 cytoplasmic staining and CD3, CD8, and CD19 membrane staining. At least 100 positive and negative cells were identified in the training set for each phenotypic marker and trained independently. Data tables were exported and merged, consolidated, and combinatorial phenotypes were analyzed in Phenoptr Reports (version 0.3.2) [20].

##### SA-β-Gal and H and E

Fresh tissue biopsies were placed in O.C.T. and stored at −80 °C until processing. Prior to staining, frozen tissue was sectioned at 5 µm in a cryostat by the Pathology Shared Resource.

Following the manufacture’s protocol (Cell Signaling Technology, #9860), senescence associated β-Galactosidase (SA-β-Gal) was assessed. Slides were fixed for 5 min, washed three times with 1X PBS and stained overnight at 37 °C. Images were obtained at 20× magnification on an Olympus IX83 microscope. H and E was performed by the Pathology Shared Resource on additional 5 µm sections.

### 2.4. Statistical Methods

The total planned enrollment across the expansion cohorts was approximately 30 patients, as this sample size provides a reasonable chance (>75%) of observing at least one or more adverse events when the true frequency of the adverse event is between 10 and 15% at the RP2D. Anticipating approximately 20 of these patients would be eligible for response evaluation, assessment of the response rate in 20 patients excludes, with 95% confidence, a true response rate of 15% or higher if no response is observed.

Adverse events were tabulated by type and grade, then summarized across both cohorts. Analysis of efficacy measures was descriptive, with best overall response summarized using the number and percent of patients in each tumor response category. Time on treatment was defined as the time from the first day of study treatment to the end of treatment visit.

All correlative analyses are hypothesis-generating and descriptive in manner. Gross lesion volumes (cm^3^) from T2wMRI and ADC values (mm^2^/s) from DWI-generated ADC maps were calculated at each timepoint using the built-in software (Siemens syngo MR [21], versions VE11C and VB20), by an MR image analyst with 18 years of experience who was blinded to the clinical information. The tumor volumes were calculated by placing a hand-drawn region of interests (ROI) over the liver lesion on each axial slice and multiplying the summed ROIs by the anatomical slice thickness (0.5 cm). The ADC values were calculated using a mono-exponential model, with the ROI placed hand-free on each axial DWI slice. ADC values from all slices of the target lesion were averaged as the ADC_mean_.

Baseline assessments of biologic markers evaluated in biopsy samples were correlated with clinical outcomes, and dynamic change in markers from baseline to C1D7 and C2D7 in both Expansion Cohort Groups were assessed using Prism version 7.0.

## 3. Results

### 3.1. Clinical Outcomes

#### 3.1.1. Solid Tumor Expansion Cohort

A total of 20 patients with refractory cancers were treated in the dose expansion portion of this trial. Demographic profiles were similar in the alisertib lead-in group (patients A-1 through A-10) and sapanisertib lead-in group (patients S-1 through S-10), with an overall median age of 60, and the majority of patients identified as non-Hispanic white and female sex. Represented tumor types in this cohort include breast adenocarcinoma (nine patients, seven HR+/HER2−, two TNBC), colorectal adenocarcinoma (four), pancreatic adenocarcinoma (three), ovarian serous carcinoma (two), renal cell carcinoma (one), and uterine serous carcinoma (one), with majority breast cancer in both groups. Patients in both groups were heavily pre-treated, with an overall median of 4 prior lines of therapy and range of 2–14 prior lines. All participants had a performance status of 0–1 at time of screening (Table 1).

Median time on study was 9 weeks across both treatment groups (range 1.1–47). In the alisertib lead-in group, the median time for treatment was 11.6 weeks (range 3.7–47), and in the sapanisertib lead-in group the median was 6 weeks (range 1.1–10) (Figure 2A and Appendix A).

The majority of patients across both lead-in groups (65%) discontinued study treatment due to disease progression. This included 80% of patients in the alisertib lead-in group. Two patients (20%) in this group discontinued study treatment due to treatment-related toxicity after one and three cycles, respectively. In the sapanisertib lead-in group, 50% of patients discontinued study due to progressive disease by RECIST 1.1, while an additional 30% discontinued prior to restaging imaging due to clinical progression. One patient in this group discontinued treatment after three cycles due to treatment-related toxicity, and another due to unrelated medical comorbidities.

The overall response rate in the 16 evaluable patients across both lead-in groups was 6%, with a stable disease rate of 38%, and disease control rate of 44%. Of the 10 patients evaluable for response in the alisertib lead-in group, the rate of partial response was 10% and the rate of stable disease was 50% for a disease control rate of 60% and disease progression rate of 40%. The one patient with partial response in the alisertib lead-in group had HR+/HER2− breast cancer and partial response to treatment with 30% decrease by RECIST 1.1. The patient discontinued treatment after 12 weeks due to toxicity related to rash. An additional patient with pancreatic cancer in this lead-in group had a 16% decrease in disease and continued on study for a total of 47 weeks. No complete or partial responses were documented in the six evaluable patients in the sapanisertib lead-in group, and 17% of patients experienced stable disease, while 83% experienced progressive disease as the best response to treatment (Figure 2B and Appendix A).

#### 3.1.2. Pancreatic Cancer Expansion Cohort

Eleven patients with pancreatic cancer were enrolled in the pancreatic cancer cohort. The median age in this group was 56, and the majority identified as non-Hispanic white and male sex. The median prior lines of therapy for metastatic disease were 2, with a range of 1–3, and all patients had a baseline ECOG performance status of 0 or 1 (Table 1).

The median time on study in this cohort was 9 weeks (range 4–27.7) (Figure 2A and Appendix A). Two patients in this cohort discontinued prior to restaging imaging due to clinical progression, while two patients discontinued treatment after 1 cycle due to toxicity, and one patient discontinued due to patient choice. Of the remaining six patients in this group evaluable for response, 67% had a documented best response of stable disease, while 33% had progressive disease. No complete or partial responses were documented (Figure 2B and Appendix A).

### 3.2. Toxicities

All patients experienced at least one adverse event (AE) while on study. Treatment related adverse events were reported in 90% of patients in the alisertib lead-in group, 70% of patients in the sapanisertib lead-in group, and 90% of patients in the pancreatic cancer expansion group. The majority of adverse events deemed related to study treatment were attributed to both drugs (48%), while 27% were attributed to alisertib alone, and 26% were attributed to sapanisertib alone.

The most common treatment-emergent toxicities reported across all treatment groups were fatigue (58%), diarrhea (42%), nausea (42%), abdominal pain (32%), and mucositis (29%) (Table 2). The most common treatment-related adverse events were fatigue (42%), mucositis (29%), hyperglycemia (26%), and nausea (23%) (Appendix A). The majority of adverse events were mild, though neutropenia was more often higher grade. This pattern of toxicity was overall similar to that documented in the phase 1 dose escalation study evaluating this combination [16].

### 3.3. Correlative Studies

#### 3.3.1. Functional Imaging

Eight patients in the solid tumor expansion cohort with liver metastases were evaluated by T2w MRI and DWI at the three specified timepoints. Six of these patients were in the alisertib lead-in group (A-2, A-4, A-5, A-7, A-8, A-9) and two (S-2, S-4) were in the sapanisertib lead-in group. In all patients, regardless of treatment group, an increase in total tumor lesion volume by T2w MRI correlated with a respective low ADC_mean_ value in DWI (Figure 3), suggesting a poor response to the treatment. Similarly, a decrease in gross lesion volume corresponded to an increase in ADC_mean_ value. Median ADC values were overall higher in patients with stable or decreased median lesion volumes on treatment as compared to those with increased lesion volumes on treatment. Clinically, low ADC values are indicative of high cellularity tissues, and increasing ADC values are associated with tumor responsiveness to chemo- and targeted therapies [22,23]. In our study, the decreasing median lesion volumes on C2D7 correlated with the increasing ADC values seen even in C1D7, which remained relatively stable near levels of background normal liver tissues, in the range of 1.43–1.66 × 10^−3^ mm^2^/s, by C2D7 (Figure 3).

Of the eight patients with functional imaging assessments, two (A-2, S-4) had stable disease per RECIST as best response on study, and one (A-5) had partial response per RECIST. All three of these patients had decreasing median lesion volumes and increasing median ADC values at these early timepoints. In one patient (S-4), visible disease at baseline became unmeasurable by imaging at C2D7. One patient (A-2) continued study for five treatment cycles; the other two discontinued study treatment for toxicity rather than progression (Figure 3C). The remaining five patients (A-4, A-7, A-8, A-9, S-2) experienced disease progression as best RECIST response. Median lesion volumes and ADC values were variable at early timepoints in these patients (Figure 3B,C).

#### 3.3.2. Tumor Tissue Analysis

A total of four patients in the alisertib lead-in group and three patients in the sapanisertib lead-in group had viable tissue for correlative analysis across the three specified timepoints. In the alisertib lead-in group, represented tumor types included HR+/HER2− breast cancer (A-2 and A-9), ovarian cancer (A-3), and colorectal cancer (A-4). One of the two patients with HR+/HER2− breast cancer had a best response of stable disease to treatment (A-2), while the other three patients in this group had best response of progressive disease. In the sapanisertib lead-in group, samples were evaluated from patients with uterine cancer (S-1), colorectal cancer (S-2), HR+/HER2− breast cancer (S-4), and triple-negative breast cancer (S-5). Best response in this group was stable disease in the patient with HR+/HER2− breast cancer (S-4), while others had progressive disease.

Across all patients in both lead-in groups, Ki67 was decreased in tumor tissues from baseline following the respective single-agent lead-in, and this decrease was maintained following combination therapy, indicating decreased cellular proliferation with the treatment, regardless of agent sequencing. A difference in caspase positive tissues was noted between the two lead-in groups, with stable or lower caspase levels as a marker of apoptosis observed in patients in the alisertib lead-in group, as compared to an overall increase in the patients in the sapanisertib lead-in group. The exception to this trend in the alisertib lead-in group was a patient with HR+/HER2− breast cancer with stable disease on treatment (A-2), for whom an increase in caspase to the highest level noted across both groups was observed after combination therapy. Similarly, the one patient with stable disease in the sapanisertib lead-in group (also HR+/HER2− breast cancer) (S-4) had significant increase in caspase with combination therapy as compared to single-agent sapanisertib (Figure 4A and Appendix A).

Decreased apoptosis in tissues from alisertib lead-in patients is consistent with prior work indicating cellular senescence rather than apoptosis as a mechanism of resistance to alisertib therapy [13,24]. Interestingly, overall higher levels of p21, a potential marker of senescence, were seen in tumor tissues from patients in the sapanisertib lead-in group as compared to the alisertib lead-in group. The importance of p53 in inducing apoptosis rather than senescence in response to alisertib therapy has been previously described [24], and as such this marker was also evaluated in the tissues. Tumor cells positive for p53 were decreased from baseline in all patients following treatment with combination therapy, and in most patients with single-agent lead-in of either agent (Figure 4A and Appendix A).

To better assess senescence in these tumor tissues, SA-β-Gal testing was performed at each treatment timepoint. In patients in the alisertib lead-in group, an initial increase in SA-β-Gal staining consistent with an increase in senescent cells was seen, followed by a decrease when sapanisertib was added in the two patients with liver as the biopsied site (A-2 and A-4). The effect was less clear in the patient in this group with lymph node biopsy (A-9). In patients in the sapanisertib lead-in group, a similar pattern of increase in senescence was also seen in the one patient with response following sapanisertib alone, with a decrease with combination therapy (S-4). This was not observed in the other patients in this group (Figure 4B).

Cyclin B1 as a marker of cell cycle activation decreased in tumor tissues across all patients and timepoints with a few exceptions. However, in the one patient in each group with stable disease, Cyclin B1 increased with combination therapy after an initial decrease in following monotherapy (A-2 and S-4) (Figure 4A and Appendix A). A decrease in mitotic division as evaluated by phospho-histone-H3 (pHH3) was demonstrated in all biopsy samples following combination treatment, even when an increase was first observed following treatment with either single-agent.

As expected, patients treated with the sapanisertib lead-in were noted to show a decrease in phosphorylated S6 (Ser235/236) as a marker of mTOR pathway activation across all timepoints. In patients in the alisertib lead-in group, the addition of sapanisertib to alisertib led to a decrease in this marker in tissues as compared to treatment with alisertib alone (Figure 4A).

Interestingly, an increase in CD45+CK- cells consistent with tumor-infiltrating immune cells were observed in tumor tissues only in patients treated with the sapanisertib lead-in, and further increased with combination therapy in two of these patients, one who experienced progression as best response and one with stable disease (S-2 and S-4) (Figure 4A). Additional testing to further evaluate the make-up of this immune cell population was performed in these patients in the sapanisertib lead-in group, with comparison to two patients in the alisertib lead-in group (A-2 and A-8). For all four patients, the tumor tissue evaluated was a metastatic liver lesion, and all four patients also had an assessment of liver metastases by functional imaging. One patient each in the alisertib lead-in group (A-2) and the sapanisertib lead-in group (S-4) had HR+/HER2− breast cancer with best response of stable disease. The other evaluated patients in the alisertib lead-in group (A-8) and sapanisertib lead-in group (S-2) had colorectal cancer with best response of disease progression. In tumor tissues of all four patients, an increase in CD56+ natural killer cells was observed following treatment with both drugs (Appendix A).

In one of the patients from each of the lead-in groups (A-2 and S-2), increases in CD19+ tumor-infiltrating B cells, CD68+ tumor-associated macrophages, CD4+ and CD8+ T cells (CD3+), as well as FOXP3+ regulatory T cells, were also observed. A few notable differences in the pattern of immune cell infiltration were seen between these two patients. In the patient in the alisertib lead-in group (A-2) who experienced stable disease on study treatment, a significant increase in both B cells and CD4+ T cells was seen with alisertib alone, followed by a decrease with addition of sapanisertib, while CD8+ T cell infiltration was relatively stable across these treatments. In the patient in the sapanisertib lead-in group (S-2) with progressive disease on study treatment, a significant further increase in both B cells and CD4+ T cells was observed with combination therapy as compared to single-agent sapanisertib. However, CD8+ T cell infiltration increased with sapanisertib alone, followed by a significant decrease with combination therapy (Appendix A).

## 4. Discussion

In an expansion cohort of patients treated with the combination of alisertib and sapanisertib at the previously defined MTD, the toxicity profile was as expected, and AEs were generally mild in severity. Overall clinical benefit was modest in this expansion cohort, with best response of partial response in a single patient in the alisertib lead-in group with HR+/HER2− breast cancer. An additional patient in this same group with pancreatic cancer experienced prolonged stable disease of approximately 11 months. Further expansion in a cohort of patients with refractory pancreatic adenocarcinoma was performed, with less compelling results, though one patient in this cohort experienced an extended period of stable disease of over 6 months, which was the second longest time on treatment across all study groups.

Both examples of prolonged stable disease are highly clinically meaningful in patients with refractory pancreatic cancer and suggest that there may be a subset of these patients who are more likely to benefit from combination therapy with alisertib and sapanisertib. Multiple systems of molecular subtyping for pancreatic cancer have been proposed according to gene expression profile and transcriptional network analysis. Among the subsets defined across these classification systems are those enriched for cell cycle effectors, including *TP53*, as well as immune pathways [25]. When considered in the context of the variability in apoptosis and immune activation noted across patients evaluated in the correlative studies in this work, these described subsets may be relevant.

Variability in response to treatment in this study was not confined to patients in the pancreatic cancer expansion cohort. Of the various tumor types represented in the expansion cohorts, multiple patients with HR+/HER2− breast cancer (four out of seven) had a median time on treatment at or beyond the median. This is in addition to a patient with HR+/HER2− breast cancer who remained on alisertib and sapanisertib for nearly 11 months when treated below the RP2D of the combination in the dose-escalation portion of this trial [16]. In a phase 2 randomized trial evaluating alisertib with or without estrogen receptor antagonist fulvestrant in endocrine-resistant HR+/HER2− metastatic breast cancer, notable clinical benefit was observed in both combination therapy as well as alisertib alone. Interestingly, this clinical activity included patients who had received prior therapy with mTOR inhibitor everolimus [26]. Taken together, these data indicate that HR+/HER2− breast cancer remains an area of interest for clinical evaluation of Aurora kinase combination therapies.

Results of correlative studies performed in expansion cohort patients, with a particular focus on differences in the two lead-in groups, help to confirm the underlying mechanisms that may contribute to some of the clinical variability noted. One such example is found in the differing levels of tumor tissue caspase as a marker of apoptosis. Decreased levels were noted in all patients in the alisertib lead-in group following treatment with alisertib alone and remained low in all but one patient following combination therapy. This is in contrast to caspase levels in the sapanisertib lead-in group, which were increased in most cases. This is consistent with treatment-induced senescence in response to alisertib [13,24], and is further supported by results of SA-β-Gal testing. Interestingly, the three patients treated with a lead-in of alisertib who had a persistent decrease in caspase in tumor tissues despite the addition of mTOR pathway activation experienced disease progression as best response to treatment. However, the one patient in this group (A-2, HR+/HER2− breast cancer) who had a notable increase in the apoptosis marker following addition of sapanisertib had stable disease as best response. This is consistent with prior evidence of mTOR pathway activation as a mechanism of resistance—de novo or acquired—to alisertib, and the potential to overcome it with mTOR inhibition [16]. That the addition of sapanisertib did not lead to this same response across all patients raises questions about the additional factors that may contribute to resistance in this case. In prior pre-clinical work, we have shown that p53 as well as p73 mediate sensitivity to aurora kinase inhibitors in mutant p53 knockdown models of TNBC [24]. In this study, p53 decreased across the majority of patients and timepoints in both lead-in groups, without clear correlation to trends in apoptosis markers. However, the effect of clinically relevant p53 mutations on such response was not evaluated and remains an area for potential further exploration.

Additional correlative results of particular interest in this study are those evaluating the immune microenvironment. Though it does not directly target the immune system, alisertib has been shown to affect the immune microenvironment in a variety of solid tumors, largely through increased infiltration of CD8+ T cells [27,28,29], which may facilitate response to treatment [28]. One proposed mechanism for this is through the release of CCR5 by senescent cells, which in turn results in increased T cell infiltration [27]. In patients in this study for whom tumor-infiltrating immune cells were evaluated in hepatic metastatic tissues, two patients had a notable increase in CD8+ T cells as compared to baseline upon initiation of single-agent therapy (A-2 and S-2). Interestingly, in the patient in the alisertib lead-in group who experienced a best response of stable disease on study (A-2), levels of infiltrating CD8+ T cells remained similar upon addition of sapanisertib. However, in the patient in the sapanisertib lead-in group who had a best response of disease progression (S-2), CD8+ T cells significantly increased with single-agent sapanisertib therapy, but then notably decreased with combination therapy. This finding is consistent with data implicating hepatic metastases as a site of disease contributing to reduced response to cancer immunotherapy by drawing CD8+ T cells from systemic circulation and into hepatic sites where they undergo apoptosis [30].

When considering functional imaging in the context of these results, it is notable that both patients with CD8+ T cell infiltration had an initial increase in ADC by functional imaging. However, the patient noted to have stable CD8+ T cells at C2D7 (A-2) had a response of stable disease by imaging following three treatment cycles, while the patient with a decline in CD8+ T cells (S-2) had progressive disease at the time of this first imaging evaluation. It may be hypothesized that the progression of disease by imaging may have occurred in the setting of reduced CD8+ T cells in the tumor microenvironment, as noted at the C2D7 timepoint. However, the other two patients with matched functional imaging and immune cell evaluation in tumor tissues did not have a change in baseline values of CD8+ T cells following either single-agent or combination therapy, and had variable responses of progressive disease, and stable disease, respectively. Though no conclusions can be drawn based on this limited correlative data, it suggests that better understanding of the immune microenvironment of hepatic metastases may be relevant to cancer therapies not considered as immune-targeted.

## 5. Conclusions

In conclusion, dual targeting of the Aurora A kinase and mTOR resulted in marginal clinical benefit in a population of patients with refractory solid tumors, including a cohort of patients with pancreatic adenocarcinoma. Within this population are represented individual patients with significant response, which may be related to the unique gene expression and transcriptional profiles of these patients that lead to variability in apoptosis and immune infiltration of tumors, among other factors. Further assessment of these characteristics, with particular consideration of the variability that may occur in the presence of hepatic metastases, should be considered in future studies of these agents.

## Figures and Tables

**Figure 1 cancers-16-01456-f001:**
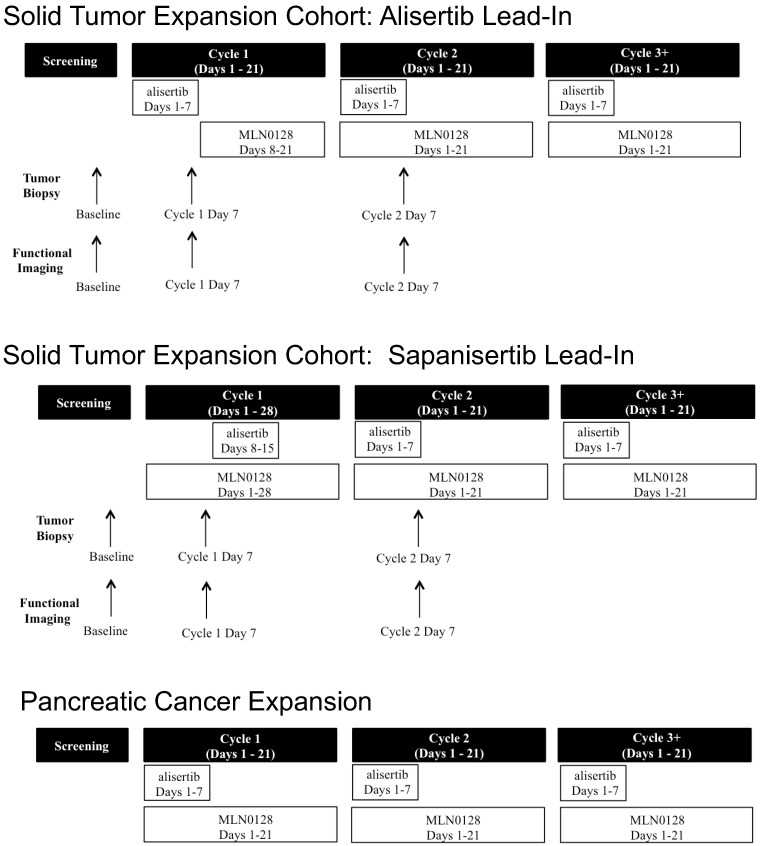
Study schema for solid tumor and pancreatic cancer expansion cohorts. Eligible patients in both cohorts were treated with the combination of alisertib 30 mg by mouth twice daily on days 1–7 in a 21-day cycle and sapanisertib 2 mg by mouth daily. In Cycle 1 of the solid tumor expansion cohort only, patients were assigned to treatment with either alisertib or sapanisertib as a single-agent on days 1–7. For the remainder of the study, these patients received combination treatment. Patients in both the alisertib and sapanisertib lead-in groups were eligible to participate in biopsy and imaging correlative studies. Biopsies were performed prior to treatment start, and again at Cycle 1 Day 7 and Cycle 2 Day 7. T2-weighted MRI was performed in this cohort at these same timepoints in patients with liver metastases only.

**Figure 2 cancers-16-01456-f002:**
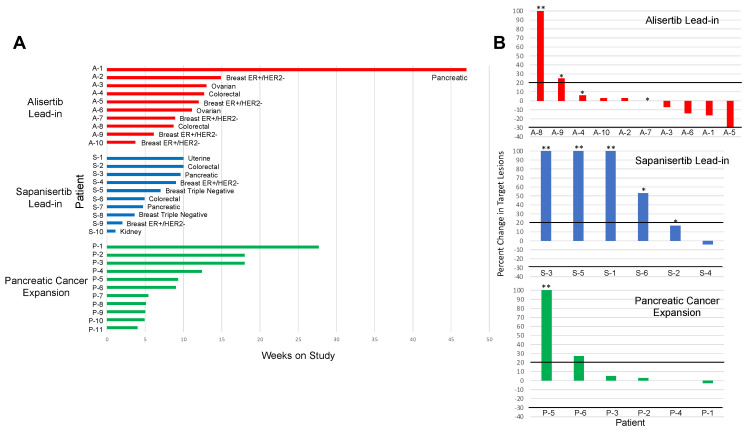
Antitumor activity of alisertib and sapanisertib in an expansion cohort of patients with refractory solid tumors. (**A**) Duration of time on study in weeks according to treatment group. Each bar represents a patient. (**B**) Best response on imaging according to treatment group in patients with evaluable disease per RECIST v1.1. * Increase in non-target lesion; ** Progression related to new lesion.

**Figure 3 cancers-16-01456-f003:**
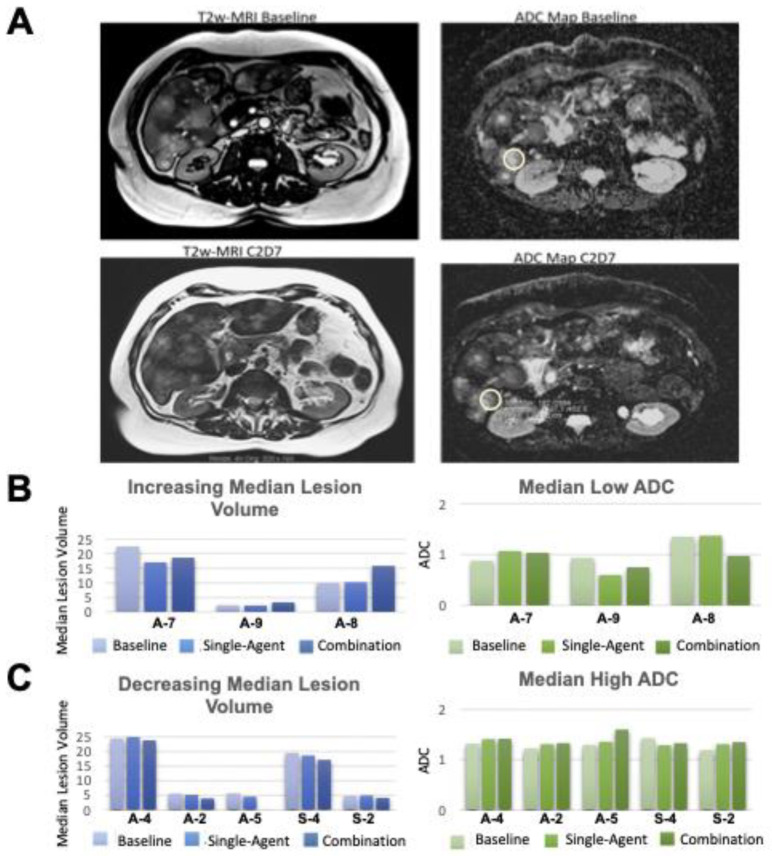
Functional imaging in select patients with hepatic metastases treated with alisertib and sapanisertib. (**A**). Representative T2w MRI and ADC maps of hepatic metastases in a patient with poor treatment response reveal increasing tumor burden and low ADC values at C2D7 of treatment; (**B**). Increased median tumor burden volume and descending ADC values between baseline, C1D7 and C2D7; versus (**C**). unchanged/decreasing tumor burden and normalized ADC values. The ADC values for normal appearing hepatic tissues were in the range 1.43 to 1.66 (×10^−3^ mm^2^/s).

**Figure 4 cancers-16-01456-f004:**
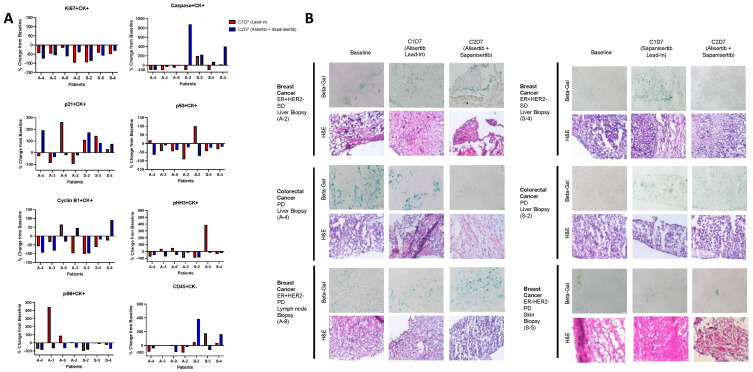
Pharmacodynamic effect of alisertib and sapanisertib as single agents followed by combination therapy in tumor tissues. (**A**) Evaluation of 8 pharmacodynamic markers as assessed by fluorescence microscopy in CK positive tissues from tumors of patients treated with single-agent lead-in of alisertib (A-2, A-3, A-4, A-9,) or sapanisertib (S-2, S-4, S-5). Percentage change from baseline was assessed in tissues from each patient following single-agent lead-in at C1D7 (red bars), and following combination therapy at C2D7 (blue bars). (**B**) Senescence associated β-Galactosidase staining in tissues of patients from both single-agent lead-in groups, with comparison to H and E staining. Images were obtained at 20× magnification on an Olympus IX83 microscope.

**Table 1 cancers-16-01456-t001:** Patient demographics and baseline characteristics.

	Number of Patients (%)		
Characteristic	Alisertib Lead-In (n = 10) Patients A-1 through A-10	Sapanisertib Lead-In (n = 10) Patients S-1 through S-10	Pancreatic Cancer Expansion (n = 11) Patients P-1 through P-11
Age			
Median (Range)	59 (51–71)	63 (39–74)	56 (38–74)
Sex			
Male	2 (20%)	4 (40%)	9 (82%)
Female	8 (80%)	6 (60%)	2 (18%)
Race/Ethnicity			
Caucasian	9 (90%)	7 (70%)	7 (64%)
Hispanic	1 (10%)	1 (10%)	3 (27%)
African American	0 (0%)	1 (10%)	1 (9%)
Asian	0 (0%)	1 (10%)	0 (0%)
Tumor type			
Breast adenocarcinoma	5 (50%)	4 (40%)	0 (0%)
Colorectal adenocarcinoma	2 (20%)	2 (20%)	0 (0%)
Ovarian serous carcinoma	2 (20%)	1 (0%)	0 (0%)
Pancreatic adenocarcinoma	1 (10%)	2 (20%)	11 (100%)
Uterine serous carcinoma	0 (0%)	1 (10%)	0 (0%)
Renal cell carcinoma	0 (0%)	1 (10%)	0 (0%)
Baseline ECOG Performance Status			
0	2 (20%)	3 (30%)	4 (36%)
1	8 (80%)	7 (70%)	7 (64%)
Prior Lines of Therapy for Metastatic Disease			
Median (Range)	4 (3–14)	3 (2–9)	2 (1–3)
1	0 (0%)	0 (0%)	1 (9%)
2	0 (0%)	3 (30%)	7 (64%)
3	2 (20%)	3 (30%)	3 (27%)
4	4 (40%)	0 (0%)	0 (0%)
5 or more	4 (40%)	4 (40%)	0 (0%)

**Table 2 cancers-16-01456-t002:** Treatment-emergent adverse events occurring in at least 20% of patients across treatment groups.

	Alisertib Lead-In N = 10		Sapanisertib Lead-In N = 10		Pancreatic Cancer Expansion N = 11		All Treatment Groups N = 31		
Number (percent)	Grade 1/2	Grade 3/4	Grade 1/2	Grade 3/4	Grade 1/2	Grade 3/4	Grade 1/2	Grade 3/4	Total
Fatigue	3	1	7	0	7	0	17 (55%)	1 (3%)	18 (58%)
Diarrhea	3	1	2	0	6	1	11 (35%)	2 (6%)	13 (42%)
Nausea	2	0	4	0	7	0	13 (42%)	0	13 (42%)
Abdominal pain	2	1	2	0	3	2	7 (23%)	3 (10%)	10 (32%)
Mucositis	3	2	2	0	1	1	6 (19%)	3 (10%)	9 (29%)
Hyperglycemia	2	1	2	0	3	0	7 (23%)	1 (3%)	8 (26%)
Anorexia	2	0	2	0	4	0	8 (26%)	0	8 (26%)
Hypokalemia	2	0	0	0	4	1	6 (19%)	1 (3%)	7 (23%)
Neutropenia	0	1	2	0	0	3	2 (6%)	4 (13%)	6 (20%)
Cognitive disturbance	0	0	2	1	3	0	5 (16%)	1 (3%)	6 (20%)
Dyspnea	3	0	0	1	2	0	5 (16%)	1 (3%)	6 (20%)

## Data Availability

Data not included in the manuscript are not provided in order to respect patient privacy. Additional details available on request from the corresponding author.

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
