# Peer review of "A Phase Ib Expansion Cohort Evaluating Aurora A Kinase Inhibitor Alisertib and Dual TORC1/2 Inhibitor Sapanisertib in Patients with Advanced Solid Tumors"

_cancers, 2024, doi:10.3390/cancers16081456_

Round 1

Reviewer 1 Report

Comments and Suggestions for Authors

Here, the authors report their results on a novel combination therapy to treat patients with refractory solid tumours.

In the already small cohort, many patients had to be discontinued due to disease progression or treatment-related toxicity. Nevertheless, results from this study can be helpful to the scientific community and clinicians in designing of future therapies.

Major comments:

I found it difficult at times to follow the study, as it is not always possible to trace each patient precisely. For example, why not use the numbering system as seen in figure 3B-C throughout the manuscript? This approach would enhance the value of Figure 2.

Also, figure 2A would benefit from more details; for example, it would be useful to know which exact patients were excluded in each group.

For the sake of reproducibility, it is important that the authors provide all antibody catalogue numbers and a brief protocol of the staining method (section 2.3.2.1).

Representatives pictures illustrating figure 4A should be added. Also, the detailed procedure for the quantification should be provided.

Minor comments:

gH2AX and p4EBP1 staining appear to have been performed, yet are neither described nor discussed. However, data on gH2AX could be of great interest.

Graph legends (y-axis) are missing throughout the manuscript. Also, bar legend in figure 3B-C (Pre, C1D7 and C2D7) could be improved.

Line 20 (simple summary): the statement “This combination was found to be tolerable” is misleading, as many patients had to be discontinued due to disease progression or treatment-related toxicity. That part of the sentence should be modulated or removed.

Lines 406-407, where it states “to have decrease of S6 as a marker”, the authors should specify that they are describing the phosphorylation status of S6, and also indicate which phosphorylated residue(s) they are examining.

Reviewer 2 Report

Comments and Suggestions for Authors

PI3K/AKT/mTOR pathway is well characterized for therapeutic resistance. A number of pathological variants have been identified in different cancer including TNBC. Drug that can target these pathological variants is well required. 

This is a good manuscript, still I have few comments: 

1.  Reason for forming two cohorts, the solid tumor expansion cohort or the pancreatic cancer expansion cohort

2. Please further elaborate breast cancer patients; ER/PR/Her-2 status.

3.Please provide the pathological features of patients. 

4. Did author also screened the PI3K/AKT/mTOR and PTEN mutation status in patients ?.

5. Author should summarize the results in tabular form, since there are different cancer types. 

Round 2

Reviewer 1 Report

Comments and Suggestions for Authors

The authors have satisfactorily addressed most of my remarks and improved the quality of the manuscript, even though I still think that graphs in Figure 3 could be improved.

Nevertheless, I have no further comments about the revised manuscript, and I support the publication with the current version.